# Spanish Fuet Sausages Fat-Reduced to Diminish Boar Taint: Sensory and Technological Quality

**DOI:** 10.3390/ani13050912

**Published:** 2023-03-02

**Authors:** Irene Peñaranda, Macarena Egea, Daniel Álvarez, María Dolores Garrido, María Belén Linares

**Affiliations:** Department of Food Science and Technology, Veterinary Faculty, University of Murcia, Espinardo, 30110 Murcia, Spain

**Keywords:** reduced-fat, dietary fibres, fuet sausage, boar taint, entire male pork

## Abstract

**Simple Summary:**

Regarding the European Union Recommendation on alternatives to surgical castration of pigs, it would be appropriate to search for new alternatives to commercialize the entire pig that could be affected by boar taint, as a sensory defect. The goal of this study was to evaluate if fat reduction in meat products could diminish the boar taint in dry-cured sausages with high levels of AND and SKA, and to study the technological and sensory quality by using inulin, β-glucan and grape fibre as fat substitutes in fat-reduced products. The addition of fibres in reduced-fat fuet provided a similar technological and sensory profile to the traditional, achieving a reduction of sexual odour. Thus, these alternative formulations in reduced-fat meat products could be useful to allow high boar-tainted pork a place in the market.

**Abstract:**

Reduced-fat cured sausages were evaluated as a strategy to reduce boar taint in entire male pork products with high levels of androstenone and skatole, both lipophilic. Three fuet-type sausages (two replicates each) were developed: the control (C) (60% lean, 33.69% fat), and two reduced-fat (60% lean 21.19% fat) products; R1, 6% inulin, 0.5% β-glucan and R2, 3% inulin, 0.5% grape skin and 1% β-glucan. All of them were prepared from entire male pork with an androstenone concentration of 6.887 µg/g and 0.520 µg/g of skatole. Significant differences (*p* ≤ 0.001) in the moisture content were described between the fuet R1 and the C and R2, which obtained the highest percentage. Regarding the CIELAB, the C samples had the highest L* value, while the R2 sausages were the darkest. Boar taint was reduced in both R1 and R2, with a greater reduction in R2 (*p* ≤ 0.000). The addition of inulin and β-glucan in fuet R1 provided a similar technological and sensory profile to C. However, both strategies provided a reduction of sexual odour, which was higher when grape skins were included. In addition, R2 gave a characteristic sausage with more odour and flavour, dark colour and overall rating than C and R1.

## 1. Introduction

The EU animal welfare recommendations to end the surgical castration of piglets without anaesthesia and/or analgesia [1] have increased the production of entire male pigs, as it is seen as the most cost-effective alternative due to its better feed-to-gain ratio and healthiness (as this meat contains a higher percentage of lean meat) [2]. However, the rearing of entire male pigs remains problematic, due to the possible occurrence of boar taint, so castration of the entire male pig has traditionally been carried out to control this undesirable odour [3]. Boar taint is a disagreeable odour and flavour that arises mainly through the accumulation of androstenone (AND: 5α-androst-16-3one) and skatole (SKA: 3-methylindole) in fatty tissue. AND is a male sex pheromone with an odour similar to urine or sweat that is perceived by approximately 40–50% of the meat consumers, while SKA is a metabolite of tryptophan associated with faecal odour or naphthalene, and is recognised by 99% of meat consumers [4]. Therefore, it is necessary to search for alternatives to reduce the boar taint in this meat, and to guarantee its commercialisation in both national and international markets. The use of spices, curing, fermentation, and smoking have been shown to reduce the boar taint of entire male pork [4,5,6,7,8,9]. However, even with the combination of several of these techniques, it is complicated to obtain entire male pork products under high-quality standards if high levels of AND and SKA are found in the meat. Cured sausages made from this meat are darker, harder, dryer, and saltier, due to the lower fat content [2], and are traditionally produced with high percentages of fat (20–30%), which leads to the presence of high concentrations of the compounds responsible for sexual odour (AND and SKA), as they have a lipophilic character [10]. As a consequence, effective technological strategies are needed to process such meat and obtain products that meet sensory quality standards, since fat from entire male pigs could be a handicap in the processing due to the main unsaturated profile. Although the production of low-fat meat products could contribute to reducing boar taint perception, it comes with some technological drawbacks, such as reduced performance, juiciness, and cured flavour, or increased toughness and therefore lower consumer acceptance [11]. 

In recent years, some strategies have been studied to produce healthier meat products without affecting their technological and sensory properties. Replacing pork fat with vegetable fibres or functional ingredients is a common practice in the meat industry, as it has shown a positive effect on the texture of low-fat meat products [11]. However, its effect on the boar taint perception in fuet sausages, a traditional Spanish meat product, is unknown. Among the fat substitutes utilised, inulin, a soluble fructooligosaccharide (FOS), is considered as a prebiotic agent due to its functional effects on the gastrointestinal microbiota [12], and is often used in numerous food formulations (dairy, meat, beverages, cereals, etc.) due to its lower caloric content [13]. It also improves organoleptic properties, water retention capacity and emulsion stability, and provides a fat-like colour [14]. β-glucan is a glucose polymer with prebiotic properties that improve the stability of emulsions by increasing the viscosity of formulations [15,16]. The use of other plant fibres such as wheat by-products [17] or fruit fibres [16,18], has also been shown to improve the technological properties of meat products. In general, these dietary fibres are mainly non-digestible polysaccharides that can contribute towards the improvement of the appearance and texture of meat products, as they slow down the drying rate, thus reducing moisture loss and providing more juiciness [19]. However, there are no studies in the bibliography consulted on the use of vegetable fibres for fat reduction in dry-cured sausages made from entire male pork, as the use of these ingredients could result in the loss of the characteristic flavour derived from fat [20]. Therefore, the objective of the present study was to assess if the fat reduction could diminish the boar taint in dry-cured sausages made from entire male pork with high levels of AND and SKA, and to study the technological and sensory quality by using inulin, β-glucan and grape fibre as fat substitutes in reduced-fat products.

## 2. Materials and Methods

### 2.1. Ingredients

The experimental procedures used in this study complied with the EU guidelines for the care and handling of research animals [21]. Thirty-one carcasses of entire male pigs with high boar taint levels were selected with human nose (trained panel) methodology [3], from three hundred animals in a slaughterhouse located in Catalonia (Spain). Subcutaneous fat from the dorsal neck region of these carcasses was analysed for androstenone (AND) with gas chromatography–mass spectrometry [22], while the presence of skatole (SKA) was analysed with high performance liquid chromatography [23]. Once analysed, pig meat from the carcasses with the highest level of boar taint; 6.887 µg/g AND/0.520 µg/g SKA, was selected. Inulin (Guinama, Barcelona, Spain) and β-glucan (Guanjie Biotech, Shaanxi, China) were used in powder concentrate form (Inulin at 3% or 6% level (*w*/*w*)/β-glucan at 0.5% or 1%), to reach the desired final content in sausage production. The red grape extract was obtained from a local purveyor (Cimusa-Dallant, SA, Murcia, Spain) and incorporated into the formulation at a concentration of 0.5% (*w/w*), or not used (Table 1). The rest of the ingredients used during fuet manufacturing were salt (Aliada, Madrid, Spain), spice mix blend (Catalina Food Solutions SL, SCM-074-SCH, Murcia, Spain; including salt, potato starch, soy protein, spices, spice extract, dextrose, sugar, lactose, sodium triphosphate (E-451), sodium ascorbate (E-301), potassium nitrate (E-252), sodium nitrite (E-250), carmine (E-120)), and water. All the ingredients of the fuet composition were kept constant in each formulation and replicated.

### 2.2. Fuet Preparation

Fuet sausages were produced in the pilot plant of the Faculty of Veterinary Medicine (University of Murcia). Lean pork ham and fat with a proximate composition of lean meat (1.25% fat; *w/w*) and fat (71.98% fat; *w/w*) from the selected carcasses of entire male pigs were used for the elaboration of three formulations of fuet-type sausages [24]. A control formulation with a regular fat content (60% lean ham/25% fat) and two reduced fat productions, R1 and R2 (60% lean and 16% fat), including two different proportions of dietary fibre (inulin, β-glucan, and grape extract), which were added separately to these formulations, were created, as shown in Table 1. An overall control formulation without extracts was also prepared to obtain a total of three different formulations of fuet sausages, with the production of two replicates of each formulation on independent days, within one week of each other. Meat and backfat were processed under refrigerated conditions for removing excess fat and connective tissues. Pork meat and fat were separately ground twice (Mainca, Barcelona PM98, III, 380 V, 3Hp, 50 Hz) through two orifice plates (meat; 25 and 10 mm ϕ, fat; 10 and 3 mm ϕ) in a meat grinder, weighed (Tanita BSE-860), packed into bags (vacuum bags, 350*550, 140 my, La Bolsera Murciana, Murcia), vacuum sealed with a packaging machine (INEI, Barcelona, Model 500), and frozen at −18 °C until product formulation, within one month. Before fuet processing, frozen meat and backfat were thawed overnight in a refrigerator at 4 °C. The appropriate amount of ingredients for each formulation (lean meat, fat, inulin, β-glucan, grape skin, salt, commercial mix, and water), were placed into a bowl and manually mixed until a highly homogeneous mass was obtained (approximately 5 min). The mixed mass was then stored in the refrigerator (4 °C) for 1 h until the mixture was compact. The refrigerated mass (approximately 4 kg) was transferred into a hand stuffer (Garhe S.A., Vizcaya, Spain), and then stuffed into artificial collagen casings 45 mm in diameter (Edicas, Salamanca, Spain), previously soaked in water at 4 °C. The fuet sausages were manually tied into approximately 15 cm links, washed to remove materials on the outside of the casing, weighed for subsequent weight control, and hung in a drying chamber for 7 days at 4 °C-80% RH, and another 3 days at 22 °C-84% RH, until the curing process of the product was completed (Figure 1) and an optimum point of moisture was reached (% moisture of fuet control: 43.6%, R1: 45.63% and R2: 45.71%; Aw control: 0.618, R1: 0.646, R2: 0.607). Finally, fuets were weighed, vacuum packed, and frozen at −18 °C in a freezing chamber until analysis, within three months, to prevent excessive dryness. 

### 2.3. Colour Measurement

Colour coordinates CIE L*a*b* were measured using a CR-400 Chroma Meter (Minolta Ltd., Milton Keynes, United Kingdom) calibrated against a standard white tile (8 mm diameter aperture, d/0 illumination system, illuminant D65, and a 2° standard observer angle). L* (lightness), a* (redness), and b* (yellowness) were measured on the cutting surface from three randomly chosen spots of three slices from each fuet sausage [16]. 

### 2.4. Moisture and Fat Determination

The determination of moisture content in the meat product samples was carried out following the process in [25], and the intramuscular fat content was established according to the process in [26], using petroleum ether (40–60 °C) as the solvent in a Soxhlet extraction equipment. Moisture and fat contents were measured in duplicate for each type of fuet.

### 2.5. Texture Profile Analysis (TPA)

The texture profile analysis (TPA) of the sausages was carried out using a Texturometer QTS-25 (Brookfield CNS Engineering Labs. Inc., Harlow, Essex, UK) with TexturePro CT V1.8 software [27]. The sausages were cut into pieces 2 cm in diameter × 2 cm in length to obtain cylindrical cores for TPA measurement, after conditioning at 23 °C. Core samples were compressed twice with a 10 mm diameter cylindrical probe (TA 10) provided with a 25 kg load cell, until they reached 50% of their original height. The force–time deformation curves were obtained with a crosshead speed of 2 mm/s and a trigger point of 5 g. TPA parameters; hardness (g), adhesiveness (mJ), chewiness (mJ), gumminess (g), cohesiveness, elasticity (mm), resilience (ability of the product to return to its initial position, J/m^3^), and extensibility (mm), were determined in triplicate. 

### 2.6. Sensory Analyses 

Eight panellists were selected from a panel previously trained in the sensory evaluation of meat products from entire male pork [10], and trained in AND (from 0 to 7 µg/g) and SKA (from 0 to 1.5 µg/g) detection [28]. Six theoretical–practical sessions lasting 1.5 h were held for specific training with reduced-fat products of entire and castrated (commercial product) male pigs. A quantitative descriptive analysis (QDA) test was performed using an unstructured 10-point scale (0 = not perceptible; 10 = maximum perception). The sensorial attributes for analysis were selected according to [16]; colour (colour intensity, brightness and homogeneous colour), odours (sausage odour, acid odour, boar taint and other odours), taste (acid, salty and bitter), flavours (sausage flavour, boar taint and other flavours), and texture (hardness, cohesiveness, chewiness and juiciness). Finally, an overall rating was included to assess the success of the strategies developed (0 = low rating; 10 = high rating). The panellists were also asked to provide their overall rating of the sausage samples. The sensory evaluation of the sausages was carried out according to the process in [29] in a standardised room [30] at the Food Technology Department (University of Murcia). The analyses were performed at the same hour of the day (at 12:00 h) for each session. A total of six samples per treatment (C, R1 and R2) and replicate (two batches) were tasted by each panellist. The samples were cut into 2 mm thick slices, and two slices coded with a random three-digit number per treatment were presented to each panellist, at a temperature of 23 °C. Sample presentation was balanced to account for order and carryover effects [31]. The panellists were provided with mineral water and unsalted bread between sample tastings to rinse their palates and to avoid flavour retention in the mouth.

### 2.7. Statistics 

The data were analysed with the SPSS 24 statistical software package [32]. The colour, texture and sensory parameters were analysed with a one-way analysis of variance (ANOVA), considering the effect of the different formulations (C, R1, and R2) as fixed sources of variation, and the two replicates, session and panellists as a random effect. The comparisons between means were performed using Tukey’s test, and differences were considered statistically different when *p* < 0.05.

## 3. Results and Discussion 

### 3.1. Composition and Instrumental Colour

Table 2 shows the results of the composition and instrumental colour of the different fuet sausage treatments. As observed, there were significant differences (*p* ≤ 0.001) in the moisture content between the fuet produced with inulin plus β-glucan (R1), and the control and that produced with grape skin sausage samples (C and R2), which obtained the highest percentage in this parameter. The grape-skin addition could retain water, resulting in values similar to those of the control, in agreement with other authors, who reported how fibres, gums, or starches as fat substitutes can help slow down the drying rate during maturation, thus reducing moisture loss [19]. Nutrition and health claims made in food regulation [33] indicate that fat reduction must be at least 30% as compared to a similar product in order for the sausages to be considered as “reduced fat”. Thus, although significant differences (*p* ≤ 0.000) were found between control and reduced-fat formulations, these fat reductions established were not sufficient for the R1 and R2 sausages to be considered “reduced fat” products. 

Regarding the CIELAB colour parameters, statistical differences (*p* < 0.001) were observed between the cured sausages, as the control samples had the highest L* coordinate value (48.4), while the R2 sausages were the darkest (41.8), as luminosity is directly related to the percentage of fat and water-retention capacity of meat products [34]. Similar results were reported by [35] on dry fermented sausages with a 16% inulin-gelled emulsion, showing how the addition of inulin provided meat products that were more similar to conventional ones, as compared with the use of other fat substitutes. The authors of [15] also obtained similar values using inulin and β-glucan in meat emulsions. Furthermore, Ref. [34] observed that the higher the concentration of inulin added, the higher the brightness values in this type of product. In the a* and b* colour coordinates, no significant differences were observed between R1 and the control sausages (*p* ≥ 0.05), but the comparison between these two and R2 (*p* = 0.008) showed that the latter obtained the lowest values for the a* coordinate. It has been observed that substitutes with a similar colour to fat, such as inulin, provide a similar colouring to conventionally cured sausages [20]. Therefore, the sausages that were made with inulin powder tended to have a more yellowish colour and a less blue one (higher value of b*), giving them the characteristic colour of this type of product [14]. On the other hand, a decrease in the reddish and yellowish colouring was obtained in the R2 fuet. Similar results were obtained by other authors, such as in [36], in meat emulsions where fat was partially replaced by grape seed oil at 0, 5, 10 and 15% and 2% bran fibre, or in cooked pork sausages made with 0.5, 1 and 2% grape marc powder (composed of stems, skins and pips) [37,38], and reduced in fat [16]. The main molecules involved in the colour of grape dietary fibre are flavonoids; therefore, the observed a* and b* values could be related to the anthocyanins present in grape skins, as it should be noted that the colour of anthocyanins can vary from red to blue depending on the pH value [39]. However, the antioxidant activity of phenolic compounds from grape skins could influence the colour of the meat, acting mainly on the heme group of myoglobin during the fermentation of these products, thereby helping to preserve the reddish colouring [40]. 

### 3.2. Instrumental Texture 

The results for the instrumental texture analysis are presented in Table 3, where statistically significant differences were observed between the three formulations (*p* ≤ 0.05) in all the parameters, except for hardness and cohesiveness (*p* > 0.05). It could be expected that fat reduction is accompanied by an increase in toughness and the instrumental chewiness of meat products, as the use of short-chain soluble fibres, such as inulin or wheat fibre (rich in β-glucan), forms a more compact emulsion than the protein matrix [17,19,27]. Furthermore, [41,42] showed that hardness increases with higher inulin concentration (7–7.5%). The results of the present study are in agreement with those from [11], who observed that hardness was not affected by the addition of 2% dietary fibre powder (inulin, FOS, cyclodextrin) as a fat substitute, since the higher moisture content and moisture/protein ratios could explain the differences in low-fat salami. However, other authors showed how the incorporation of konjac gel or inulin reduced hardness values in sausages [35,43]. Therefore, the effects of dietary fibres on this parameter depend on both the amount of fibre and the type (powder or gel), as gels provide softer sausages. 

The impact of dietary fibre addition on chewiness indicated that the R2 formulation had a higher value (201.2) than the control (147.76) (*p* ≤ 0.002). The increase could probably be attributed to the addition of grape fibre and the higher levels of β-glucan in this formulation, as some dietary fibres have the ability to strengthen the connections between different matrix components [44]. The control group reached the highest fat %, as fat promotes the stimulation of salivary receptors. Thus the higher the intramuscular fat content, the greater the juiciness and the lower the chewiness, as previously described [45]. Similar results were obtained by [12,44], as the addition of inulin or FOS in reduced-fat cured sausages increased chewiness. Inulin promotes harder and chewier low-fat products, but in excess of 6%, it can lead to poorer texture characteristics. However, in the present study, an acceptable texture profile was described for both reduced-fat products. Furthermore, Ref. [16] observed a lower chewiness with the addition of inulin in reduced-fat meat matrices (Spanish sausages), although other authors, such as [44], described an increase with the incorporation of gel agents such as cellulose-chitosan. 

For gumminess, R1 and R2 sausages obtained a higher value with respect to the control (*p* ≤ 0.004). Similar results were reported by other authors for gumminess in sausages and Spanish sausages reduced in fat with FOS or grape pomace, as gumminess and chewiness, being the secondary parameters of toughness, behave similarly [12,16]. With the substitution of entire pork loin fat with inulin and β-glucan, no significant differences were observed for the parameters of adhesiveness, extensibility, cohesiveness, and elasticity, with respect to the control sample (*p* > 0.05). On the other hand, the addition of grape skin reduced the adhesiveness, had no effect on cohesiveness, and increased the extensibility and elasticity of the R2 sausages (*p* ≤ 0.05). For resilience (the ability of food to absorb energy when deformed, before reaching its yield strength, and to release it when the load is released), statistically significant differences were obtained between the three formulations (*p* ≤ 0.000), with the control sausage obtaining the highest value for this parameter, due to the higher fat content, followed by R1, while the lowest value was obtained by R2. Thus, the results revealed that the fibres employed varied widely in their effects on the parameters studied. In general, the effect observed was influenced both by the type of dietary fibre used, as well as its concentration and hydrophobicity, and the interactions established between the main components of the meat matrix (fat, proteins, and hydrocolloids) [19,44]. 

Many studies have analysed the instrumental profile of dry-cured sausages, obtaining results that differed from each other, depending on the type and concentration of the fat substitute. In this sense, Ref. [35] showed an increase in the adhesiveness of sausages with the use of inulin gel instead of powder, while [19] observed how cohesion decreased as more fat was replaced by quinoa. On the other hand, Ref. [12] analysed how binary mixtures of inulin and FOS increased cohesion to resemble the control product, as described in the present study. With respect to elasticity, Ref. [27] found that sausages made with 4% starch as a fat substitute obtained higher values. However, elasticity was not affected with the addition of 3% inulin in salami [11]. These parameters are important in the handling and slicing of sausages and can make it difficult to do [41]. In addition, cured sausages made from entire male pork are tougher and drier due to the lower fat content in this meat, which causes higher processing losses [2,9]. Therefore, the maintenance of adequate textural properties is desirable. The doses used in the R2 formulation resulted in less adhesive, and more elastic and chewy sausages, with a correct cohesiveness and hardness similar to the control sample.

### 3.3. Sensory Analyses

The results of the sensory evaluation of the three fuet formulations are shown in Table 4. Statistically significant differences (*p* ≤ 0.05) were observed in all the attributes studied, except for homogeneity, acid odour/flavour, bitterness, cohesiveness, or juiciness (*p* > 0.05). The scores provided by the panellists for the appearance attributes (colour, brightness, and homogeneity) were in accordance with the results obtained in the instrumental analysis. The R1 formulation showed a similar behaviour to the control sample, as described in other studies on cured sausages reduced in fat with inulin [20,35]. In contrast, R2 obtained a higher colour score and lower brightness (*p* ≤ 0.000) than the control and R1 samples, due to the colour of the anthocyanins present in the grape skin [39] and the lower fat content of this formulation [34], in comparison with the control group. The R2 formulation obtained the highest score (6.8 and 7.0) for sausage odour and flavour, respectively, while R1 had a higher value (5.9 vs. 4.9) only for sausage flavour as compared to the control sample (*p* ≤ 0.000). The grape skin inclusion in the reduced-fat R2 fuet sausage decreased the boar-taint defect by masking unpleasant odour and flavour and highlighting the traditional organoleptic features of a dry-fermented sausage [4]. The authors of Ref. [39] described an intense fermented odour in the meat preparations with the addition of grape pomace. Furthermore, phenols and organic acids in the pomace can cause astringency, acidity, and bitterness. Our results showed slightly higher scores for the attributes of off-odour and flavour (0.2 and 0.3, respectively), and salty, in the R2 formulation (*p* ≤ 0.007), but no higher acidity than the control and R1 samples (*p* > 0.05).

Boar-taint odour and flavour were reduced in both R1 (35.4% and 41.6%) and R2 (77.6 and 80.8%) formulations, with a greater reduction in R2 (*p* ≤ 0.000). Dry-fermented products require high percentages of fat that could imply high concentrations of boar taint, because the molecules involved are lipophilic [10]. However, the dry-fermentation process can decrease boar taint, as fermentation results in changes in aroma [7], and the drying process leads to oxidation of the fat fraction [46]. Moreover, as fuet sausages are consumed cold, the boar taint is more difficult to perceive [4,9]. However, these processes are insufficient to eliminate this perception when very high levels of AND and SKA are present. Dry-cured products usually lose more than 30% of their weight during the curing process, thus increasing the absolute concentration of these compounds and their perception in the final product [47]. Previous studies have shown how the use of conventional additives such as spices, smoking [8,9], and dilution of 10% entire male pork with uncontaminated meat [48], successfully masked the boar taint in dry-fermented sausages. It was therefore to be expected that the R2 formulation would obtain the lowest scores on these attributes, as the reduction in fat, together with the addition of grape skins, which intensifies the fermented odour/flavour typical of these products [39], helped to minimize their perception. Hence, it can be concluded that fine tuning of production processes (cooking and/or smoke condensates), and mixing with non-boar tainted meat (targeted dilution), can significantly reduce the perception of boar taint, thus potentially reducing rejection by consumers [49]. 

In terms of textural attributes, the reduced-fat formulations obtained the highest scores for hardness and chewiness (*p* ≤ 0.05). As proven in the instrumental texture analysis, dietary fibres contribute to the strengthening of the bonds between the different components of the protein matrix [27,44]. For cohesiveness and juiciness, no statistically significant differences were observed (*p* > 0.05), so that the fat-reduced formulations with inulin, β-glucan and grape skin had textural characteristics similar to the control sausages. Fat reduction could affect the juiciness of the products, but the addition of the vegetable fibre seemed to be sufficient to obtain similar values as the control sample. Similar results were obtained in previous works with reduced-fat Frankfurter sausages with the addition of inulin, β-glucan, and grape pomace, as inulin has a high capacity to bind water and form juicy and stable emulsions similar to control ones [16]. In dry-fermented, reduced-fat sausages with inulin and FOS, no differences in sensory texture were observed either [11,12]. Finally, the level of masking by the strategies (overall rating) was significantly higher for R2, followed by R1 and C (*p* ≤ 0.000). The R2 formulation provided improved odour and cured flavour, as previously described in fresh and Frankfurter reduced-fat sausages with grape skins added [16].

## 4. Conclusions

The development of reduced-fat fuets (R1 and R2) under a high sensory and technological quality profile was carried out in the present study. In terms of boar-taint masking, both formulations (R1 and R2) reduced its sensory perception on the product, but R2 obtained the best masking properties by preserving the sausage odour and flavour of the traditional Spanish fuet sausages. Therefore, the incorporation of inulin, β-glucan and grape fibre, as fat substitutes in fuet production made with entire male pork with high levels of AND and SKA, could be considered as a useful technological masking strategy for boar-tainted pork. The formulation R1 (6% inulin and 0.5% β-glucan) showed a similar technological behaviour to the control sausages, while R2 (3% inulin, 1% β-glucan and 0.5% grape skin) showed the darkest product with the highest overall sensorial rating. 

## Figures and Tables

**Figure 1 animals-13-00912-f001:**
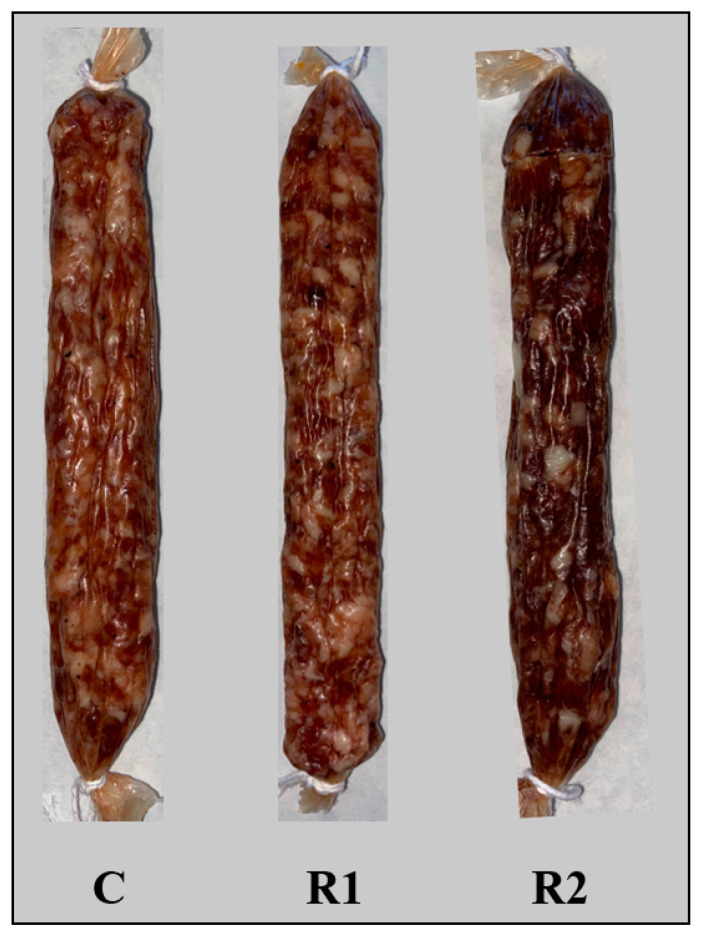
Fuet sausage and formulations. C: control with regular fat content; R1: reduced fat with inulin + β-glucan; R2: reduced fat with inulin + β-glucan + grape skin.

**Table 1 animals-13-00912-t001:** Formulations used in preparing the fuet sausages (%).

Formulation (%fat)	Lean	Fat *	Inulin	Grape Skin	β-Glucan	Salt	Mix	Water
C (25)	60	33.69				1	4	1.31
R1 (16)	60	21.19	6		0.5	1	4	7.31
R2 (16)	60	21.19	3	0.5	1	1	4	9.31

C: control fuet made with 25% fat. R1/R2: Low fat fuets made with 16% fat and different combinations of inulin, β-glucan and grape skin. * Final fat content in raw material used for fuet sausage production according to the proximal composition of meat and fat samples.

**Table 2 animals-13-00912-t002:** Composition and instrumental colour of fuet sausages (means ± SD).

Parameters	C	R1	R2	*p*-Value
Moisture (%)	21.4 ± 0.00 ^b^	19.5 ± 1.43 ^a^	21.3 ± 0.36 ^b^	0.002
Fat (% MM)	35.2 ± 2.65 ^b^	26.9 ± 1.93 ^a^	26.2 ± 1.93 ^a^	0.000
CIELAB colour				
L*	48.4 ± 1.83 ^c^	45.1 ± 1.38 ^b^	41.7 ± 1.17 ^a^	0.000
a*	11.6 ± 1.78 ^b^	11.7 ± 1.68 ^b^	9.1 ± 1.22 ^a^	0.004
b*	6.3 ± 0.53 ^b^	5.9 ± 0.94 ^ab^	5.2 ± 0.27 ^a^	0.008

C: control regular fat content; R1: reduced fat with inulin + β-glucan; R2: reduced fat with inulin + β-glucan + grape skin. L*: luminosity, a*: red-green; b*: yellow-blue. ^a–c^: Tukey’s test *p* < 0.05. Fat (% MM): fat percentage in moisture matter. Sample size (n) 18 (6 samples per type of fuet).

**Table 3 animals-13-00912-t003:** Instrumental texture of fuet sausages (means ± SD).

Parameters	C	R1	R2	*p*-Value
Hardness (g)	9418.3 ± 464.24	10,069.0 ± 1146.32	10,206.0 ± 1007.07	0.212
Gumminess (g)	2907.2 ± 140.90 ^a^	3348.7 ± 172.03 ^b^	3341.6 ± 398.36 ^b^	0.004
Chewiness (mJ)	147.8 ± 13.10 ^a^	178.8 ± 24.99 ^ab^	202.0 ± 36.04 ^b^	0.002
Adhesiveness (mJ)	4.4 ± 0.43 ^b^	3.9 ± 0.60 ^ab^	3.6 ± 0.49 ^a^	0.014
Resilience	1.5 ± 0.19 ^c^	1.4 ± 0.06 ^b^	1.2 ± 0.09 ^a^	0.000
Extensibility	21.5 ± 5.47 ^a^	18.0 ± 3.84 ^a^	29.0 ± 5.22 ^b^	0.001
Cohesiveness	0.3 ± 0.02	0.3 ± 0.02	0.3 ± 0.03	0.718
Elasticity (mm)	5.2 ± 0.29 ^a^	5.4 ± 0.76 ^ab^	5.9 ± 0.36 ^b^	0.039

C: control regular fat content; R1: reduced fat with inulin + β-glucan; R2: reduced fat with inulin + β-glucan + grape skin. ^a–c^: Tukey’s test *p* < 0.05. Sample size (n) 18 (6 samples per type of fuet).

**Table 4 animals-13-00912-t004:** Sensory analyses of fuet sausages (means ± SD).

Attributes	C	R1	R2	*p*-Value
Colour	5.6 ± 0.80 ^a^	6.4 ± 0.56 ^b^	8.3 ± 0.72 ^c^	0.000
Brightness	4.8 ± 0.70 ^b^	4.9 ± 1.01 ^b^	3.6 ± 0.88 ^a^	0.000
Homogeneity	9.9 ± 0.43	9.9 ± 0.46	9.7 ± 0.63	0.390
Sausage odour	4.9 ± 0.69 ^a^	5.7 ± 1.41 ^a^	6.8 ± 1.82 ^b^	0.000
Acid odour	0.9 ± 0.58	0.7 ± 0.76	0.8 ± 0.75	0.469
Off odour	0.0 ± 0.00 ^a^	0.0 ± 0.00 ^a^	0.2 ± 0.37 ^b^	0.007
Boar taint odour	4.3 ± 1.32 ^c^	2.8 ± 1.08 ^b^	1.0 ± 0.53 ^a^	0.000
Acid	1.1 ± 0.88	1.0 ± 0.98	1.1 ± 1.04	0.938
Salty	5.4 ± 0.46 ^a^	5.6 ± 0.44 ^ab^	5.9 ± 0.70 ^b^	0.021
Bitter	0.2 ± 0.56	0.2 ± 0.57	0.2 ± 0.39	0.986
Sausage flavour	4.9 ± 0.92 ^a^	5.9 ± 0.78 ^b^	7.0 ± 1.23 ^c^	0.000
Off flavour	0.0 ± 0.00 ^a^	0.0 ± 0.00 ^a^	0.3 ± 0.52 ^b^	0.000
Boar taint flavour	5.3 ± 1.95 ^c^	3.1 ± 1.38 ^b^	1.0 ± 0.64 ^a^	0.000
Hardness	5.7 ± 0.61 ^a^	6.0 ± 0.66 ^ab^	6.5 ± 1.15 ^b^	0.016
Cohesiveness	5.9 ± 1.63	5.8 ± 0.59	6.1 ± 0.87	0.627
Chewiness	5.3 ± 0.50 ^a^	5.9 ± 0.49 ^b^	5.9 ± 1.16 ^b^	0.016
Juiciness	5.0 ± 0.75	4.5 ± 0.93	4.6 ± 1.50	0.366
Overall rating	4.1 ± 1.57 ^a^	5.8 ± 1.19 ^b^	8.0 ± 1.58 ^c^	0.000

C: control regular fat content; R1: reduced fat with inulin + β-glucan; R2: reduced fat with inulin + β-glucan + grape skin. ^a–c^: Tukey’s test *p* < 0.05. Scores from 0—not perceptible to 10—maximum perception, an unstructured 10-point scale. Sample size (n) 36 per panellist (12 samples per type of fuet).

## Data Availability

Data supporting reported results are available upon request.

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
