# Peer review of "Spanish Fuet Sausages Fat-Reduced to Diminish Boar Taint: Sensory and Technological Quality"

_animals, 2023, doi:10.3390/ani13050912_

Round 1

Reviewer 1 Report

In the paper “Spanish Fuet sausages fat-reduced to diminish boar taint: Sensory and Technological quality” evaluate the possibility to diminish boar taint in sausage produced with entire boar male meat by using a reduced amount of fat and the addition of inulin, B-glucan and grape fiber as fat substitute. Color measurement, moisture and fat determination, texture profile analysis and sensory evaluation were carried out.

The manuscript is interesting and well written, I have just a couple of concerns.

Abstract: I would limit generic information in the abstract (e.g. lines 19-21) and focus on the experimental plans and the more relevant results. Consider including the more relevant statistical differences in the abstract.

Minor comments

Line 21: change with “technological properties”.

Delete space between line 45-46.

Line 65: when used for the first time, indicate first the word in full and after the acronym (between brackets)

line 243: add space after bullet point.

Line 388: add “carried out” or a similar verb after “was”.

Author Response

In the paper “Spanish Fuet sausages fat-reduced to diminish boar taint: Sensory and Technological quality” evaluate the possibility to diminish boar taint in sausage produced with entire boar male meat by using a reduced amount of fat and the addition of inulin, B-glucan and grape fiber as fat substitute. Color measurement, moisture and fat determination, texture profile analysis and sensory evaluation were carried out.

The manuscript is interesting and well written, I have just a couple of concerns.

Abstract: I would limit generic information in the abstract (e.g. lines 19-21) and focus on the experimental plans and the more relevant results. Consider including the more relevant statistical differences in the abstract.

Abstract has been corrected in accordance with the referee suggestions. Please see lines 17-29.

Minor comments

Line 21: change with “technological properties”.

Deleted since the abstract has been corrected.

Delete space between line 45-46.

Corrected please see line 45.

Line 65: when used for the first time, indicate first the word in full and after the acronym (between brackets)

Corrected please see line 65.

line 243: add space after bullet point.

Corrected please see line 245.

Line 388: add “carried out” or a similar verb after “was”.

Corrected please see line 391.

Reviewer 2 Report

General comments:

The manuscript deals with the evaluation of alternative ingredients (dietary fiber) for fat replacement in Spanish Fuet sausages produced from boar-tainted carcasses.  The effect on the product's instrumental and sensory quality is evaluated. In my opinion, the topic addressed in the presented manuscript is highly relevant to animal production and falls into the scope of the journal “animals”. Namely, recently a lot of attention is given to technological solutions that could mask boar taint in pork products. The authors present an innovative strategy: reducing fat content and replacement of fat with alternative ingredients.   

In general, the manuscript is well-written and organized. The study design, analytical methods, and statistical approach used are appropriate.  However, I miss some information:

- the androstenone and skatole content in the final product as some studies show that by curing process initial concentration can be reduced. Additionally in the presented results (Table 4) panelists have given significantly lower boar taint scores to R1 and R2 – R2 might have a boar taint level lower than 0.5 ppm which is considered as the threshold level. Thus it would be interesting to have the concentration of boar taint compounds in the final product.

- the information on the fatty acid composition and oxidative stability of the product (e.g. T-bars) because fat from boars contains more PUFA which is unfavorable for processing

- number of repetitions for each experimental group should be reported in the table

- number of decimal places should be reduced in tables to make them more readable

I recommend a minor revision of the manuscript in its present form and suggest the following changes.

Specific comments:

Introduction

Line 45-46: Please delete the long spacing between “reduce the” and “boar taint”

Line 54: You could add a statement that fat from boars is also problematic for processing due to unfavourable fatty acid composition, even if the carcass in low tainted (higher PUFA content is associated with higher oxidation of such product)

Line 63: The reference for the statement in lines 61-63 is missing

Materials and methods

Line 92: Please insert “from the carcasses with the” after “pig meat”

Line 95: please replace “and with “or”

Line 104: chose one unit g/100g or %

Lines 106-107: I suggest changing “Fat content used in each formulation …” to “Final fat content in row material used for Fuet sausage production according to…

Results and discussion

Lines 237, 302, and 346: Please correct the titles to have “(mean ± SD)” at the end of the title

Table 2: I suggest reducing the number of decimal points to make the table more readable; also b should be in superscript after 11.61±1.78 in a* value reported for the C group; please explain what is MM in Fat (%MM) in the footnote of the table

Table 3: I suggest reducing the number of decimal points to make the table more readable at least for hardness, gumminess, and chewiness

Line 346: Please ad “of” before “Fuet”

Line 346: Please add “but” before “with”

Author Response

The manuscript deals with the evaluation of alternative ingredients (dietary fiber) for fat replacement in Spanish Fuet sausages produced from boar-tainted carcasses.  The effect on the product's instrumental and sensory quality is evaluated. In my opinion, the topic addressed in the presented manuscript is highly relevant to animal production and falls into the scope of the journal “animals”. Namely, recently a lot of attention is given to technological solutions that could mask boar taint in pork products. The authors present an innovative strategy: reducing fat content and replacement of fat with alternative ingredients.   

In general, the manuscript is well-written and organized. The study design, analytical methods, and statistical approach used are appropriate.  However, I miss some information:

- the androstenone and skatole content in the final product as some studies show that by curing process initial concentration can be reduced. Additionally in the presented results (Table 4) panelists have given significantly lower boar taint scores to R1 and R2 – R2 might have a boar taint level lower than 0.5 ppm which is considered as the threshold level. Thus it would be interesting to have the concentration of boar taint compounds in the final product.

- the information on the fatty acid composition and oxidative stability of the product (e.g. T-bars) because fat from boars contains more PUFA which is unfavorable for processing

Unfortunately information required was not evaluated in this study. Thanks so much for this suggestion, authors will take into account for future studies.

- number of repetitions for each experimental group should be reported in the table

Included, please see tables.

- number of decimal places should be reduced in tables to make them more readable

Reduced the number of decimals in means, please see tables.

I recommend a minor revision of the manuscript in its present form and suggest the following changes.

Specific comments:

Introduction

Line 45-46: Please delete the long spacing between “reduce the” and “boar taint”

Corrected please see line 45.

Line 54: You could add a statement that fat from boars is also problematic for processing due to unfavourable fatty acid composition, even if the carcass in low tainted (higher PUFA content is associated with higher oxidation of such product)

Corrected please see lines 55-56.

Line 63: The reference for the statement in lines 61-63 is missing

Corrected please see lines 63.

Materials and methods

Line 92: Please insert “from the carcasses with the” after “pig meat”

Corrected please see lines 92.

Line 95: please replace “and with “or”

Corrected please see lines 95.

Line 104: chose one unit g/100g or %

Selected a unit, please see title Table 1

Lines 106-107: I suggest changing “Fat content used in each formulation …” to “Final fat content in row material used for Fuet sausage production according to…

Corrected please see lines 106-107.

Results and discussion

Lines 237, 302, and 346: Please correct the titles to have “(mean ± SD)” at the end of the title

Corrected please see tables.

Table 2: I suggest reducing the number of decimal points to make the table more readable; also b should be in superscript after 11.61±1.78 in a* value reported for the C group; please explain what is MM in Fat (%MM) in the footnote of the table

Corrected please see table 2.

Table 3: I suggest reducing the number of decimal points to make the table more readable at least for hardness, gumminess, and chewiness

Corrected please see table 3.

Line 346: Please ad “of” before “Fuet”

Corrected please see line 348.

Line 346: Please add “but” before “with”

Authors do not understand exactly which is the change to do in the line referred.